# Angiotensin Regulation of Vascular Homeostasis: Exploring the Role of ROS and RAS Blockers

**DOI:** 10.3390/ijms241512111

**Published:** 2023-07-28

**Authors:** Nikolaos Koumallos, Evangelia Sigala, Theodoros Milas, Nikolaos G. Baikoussis, Dimitrios Aragiannis, Skevos Sideris, Konstantinos Tsioufis

**Affiliations:** 1Cardiothoracic Department, Hippokration Hospital of Athens, 11527 Athens, Greece; evitasig@hotmail.com (E.S.); theo_milas@yahoo.gr (T.M.); nikolaos.baikoussis@gmail.com (N.G.B.); 2Cardiology Department, Hippokration Hospital of Athens, 11527 Athens, Greece; arajohn@hotmail.com (D.A.); skevos1@otenet.gr (S.S.); ktsioufis@hippocratio.gr (K.T.)

**Keywords:** redox signaling, oxidative stress, RAS, ACE inhibitors, ARBs

## Abstract

Extensive research has been conducted to elucidate and substantiate the crucial role of the Renin-Angiotensin System (RAS) in the pathogenesis of hypertension, cardiovascular disorders, and renal diseases. Furthermore, the role of oxidative stress in maintaining vascular balance has been well established. It has been observed that many of the cellular effects induced by Angiotensin II (Ang II) are facilitated by reactive oxygen species (ROS) produced by nicotinamide adenine dinucleotide phosphate (NADPH) oxidase. In this paper, we present a comprehensive overview of the role of ROS in the physiology of human blood vessels, specifically focusing on its interaction with RAS. Moreover, we delve into the mechanisms by which clinical interventions targeting RAS influence redox signaling in the vascular wall.

## 1. Introduction

Reactive oxygen species (ROS) are considered highly reactive molecules that, depending on the amount produced, can affect cells and vascular functions [1]. Their precise regulation of production in the endothelium is crucial for controlling various cell functions in the vascular system [2,3,4]. Otherwise, excessive production of ROS (including superoxide O_2_^_^, hydroxyl radical ^_^OH, and peroxynitrite anion ONOO^_^) can cause the disruption of redox homeostasis, damage the immune system response, and favour the manifestation of vascular diseases [5,6]. This situation can occur in various circumstances where there is an imbalance between ROS generation and antioxidant defence mechanisms, such as chronic inflammation, ischemia-reperfusion injury, aging, and metabolic disorders. Another important factor is impaired endothelial function (since endothelium is the main source of ROS), which disrupts the production of protective blood agents and contributes to cytotoxic effects, cell death, and, inevitably, oxidative stress [7,8]. Interestingly, it is worth noting that ROS have beneficial effects due to their cytotoxic properties and can serve as a defence mechanism against infections [9].

As for the overproduction of ROS in the development of cardiovascular disease, damage to the cardiovascular system caused by oxidative stress favours diseases such as hypertension, atherosclerosis, and heart failure [10,11]. The renin-angiotensin system (RAS) interacts with ROS and contributes to these diseases. Angiotensin II (Ang II), as a key component of the RAS, plays an important role in this interaction as it can stimulate ROS production through various mechanisms, such as activating nicotinamide adenine dinucleotide phosphate (NADPH) oxidase through its binding to the AT1 receptor (AT1R) [10]. This overproduction of ROS over Ang II can in turn affect the activity of components of the RAS, such as increase in the expression of the Angiotensin I to ACE and the production of renin. In situations where excessive ROS production plays a role in the pathogenesis of disease progression, therapeutic interventions may be beneficial [12]. Modulators of this complex interaction are angiotensin-converting enzyme inhibitors (ACEΙs) and angiotensin receptor blockers (ARBs).

In this review, we aim to explore the intricate relationship between ROS and the angiotensin system. We will delve into the role of ACEΙs and ARBs in modulating the interplay. By exploring the physiological aspects of ROS in the vasculature and their interaction with the RAS, we can improve our understanding of the mechanisms that control redox signaling within the vascular wall. This investigation will provide valuable insights into the therapeutic implications of modulating RAS for the treatment of vascular diseases and optimization of vascular homeostasis.

## 2. Discussion

### 2.1. Physiology of the RAS in Vascular Regulation

Understanding the physiology of RAS is vital for comprehending the interplay between ROS and RAS in vascular regulation. Increased levels of Ang II, the main effector of the RAS molecule, play a crucial role in cardiovascular disease progression when its production or signaling becomes dysregulated [10,11]. The physiological intricacies of the RAS have been thoroughly examined and comprehensively expounded upon in previous scientific investigations [10]. However, in the context of our present research on the influence of ROS in the progression of cardiovascular diseases, it is imperative to revisit and restate the fundamental principles of RAS physiology. By revisiting this well-established foundation, our objective is to establish a cohesive link between the RAS and the role of ROS in the development and manifestation of cardiovascular disorders.

The release of renin (an enzyme produced by conditions of reduced perfusion such as low blood pressure, low blood volume, or sympathetic stimulation), will activate a sequential processing of glycoprotein angiotensinogen (AGT), leading to the production of the decapeptide Ang I [12]. Subsequently, Ang I is further cleaved by ACE, a membrane-bound metalloprotease, and converts to Ang II. Once Ang II is formed it binds G-protein-coupled receptors (AT1-4R) [13,14,15]. AT1Rs are the prime mediators of Ang II leading to vasoconstriction, aldosterone and vasopressin secretion, and sodium and water retention. Because of this, AT1Rs play a crucial role in cardiovascular regulation, inflammation, fibrosis, endothelial dysfunction, and organ damage like nephrosclerosis [16,17] (Figure 1).

On the other hand, the effects of AT2-4Rs are not fully understood, but they act differently than AT1Rs. It is believed that AT2Rs, when activated, counteract the short- and long-term effects of AT1Rs (Figure 1) leading to beneficial antiproliferative and vasodilatory effects [18]. Other potential effects of AT2Rs include the regulation of cell growth and processing of the neuronal tissue [19]. AT3Rs were recently discovered and their exact role has not been elucidated. AT4Rs play a protective role in thrombotic effects as a fibrinolysis buffer by controlling the production of plasminogen activator inhibitor-1 (PAI-1). The nature of AT4Rs has been extensively investigated in the past with insulin-regulated aminopeptidase (IRAP) being the prominent candidate [20]. Angiotensin IV not only inhibits the peptidase activity of this enzyme but also facilitates IRAP translocation to the cell surface and enhances insulin-mediated glucose uptake [20]. In addition to the impacts mediated by G-protein-coupled receptors, recent discoveries of alternative enzyme systems and novel effector peptides have broadened our conventional understanding [21]. Among these systems are the receptors for prorenin and the G-protein-coupled receptor MAS [22]. The identification of these systems enhances our understanding of the intricate nature of the RAS by revealing that prorenin can activate renin and subsequently trigger protein kinases ERK1 and ERK2 [23,24]. The activation of these receptors introduces additional proliferative and metabolic effects that are not dependent on Ang II [25]. For blood pressure regulation, Ang II fast response to vascular changes is important. G protein-depended pathways enhance smooth muscle cell contraction [26] and trigger molecules that will contract smooth muscle cells. Moreover, when Ang II activates the G protein, it triggers the activation of phospholipase C, which leads to the production of inositol-1,4,5-trisphosphate (IP3) and diacylglycerol. These molecules then initiate the release of calcium ions (Ca^2+^) into the cytoplasm. The binding of Ca^2+^ to calmodulin activates myosin, a protein involved in muscle contraction, and enhances its interaction with actin. As a result, smooth muscle cells contract, contributing to the modulation of blood pressure.

The provided section of physiology elucidates the mechanism behind blood pressure regulation, which are mainly regulated by the rapid action of Ang II. By activating smooth muscle cell contraction through the G protein-dependent pathway, Ang II triggers a cascade of events involving phospholipase C, inositol-1,4,5-trisphosphate (IP3), diacylglycerol, and calcium ions (Ca^2+^). This series of molecular interactions ultimately leads to the contraction of smooth muscle cells. Understanding these intricate processes helps us grasp the physiological basis of blood pressure adjustments in response to changes in posture.

### 2.2. Pathological Implications of the RAS and the Involvement of ROS

As established thus far, the pathological manifestations of the RAS (abnormal cellular proliferation, inflammation, disrupted vascular balance) are closely related to the excessive presence and prolonged exposure of Ang II [27]. ROS, such as O_2_^•_^ (superoxide) and H_2_O_2_ (hydrogen peroxide), contribute to these pathological processes through Ang II signaling [28,29,30,31]. In the cardiovascular system, Ang II contributes significantly to hypertension through its central, vascular, or renal effects. The multimeric enzyme of NADPH oxidase-derived ROS production is a deleterious equation in the development of Ang II-induced hypertension. NAPDH oxidase induces Ang II-induced oxidative stress, due to increased enzymatic activity. This occurs because of the rapid translocation and phosphorylation of cytosolic subunits of the small GTPase rac1 and p47phox to the cytochrome complex, via protein kinase C (PKC) [32,33]. Thereafter, PKC activates Janus kinase (JAK), which transduces and activates the JAK/STAT signaling pathway. This sequence promotes Ang II multiplication processes. Other components in revealing Ang II adverse effects are the early growth responsive genes and redox-sensitive proteins (c-Src, epidermal growth factor EGFR) [34,35,36]. Specifically, EGFR activates the Ras/Raf/ERK cascade, which subsequently upregulates c-Fos. Together with c-Jun, activated by c-Src via JNK, c-Fos forms the transcription factor known as activator protein-1 (AP-q1) [33].

Elevated levels of Ang II can have additional consequences on cellular viability and potentially induce DNA damage. The increased presence of O_2_^•_^ and H_2_O_2_ activates additional redox-sensitive proteins, including p38/MAPK, which in turn stimulates the pro-survival factor Akt [37,38]. In a cascade of subsequent reactions, Akt inhibits various pro-apoptotic proteins. Another essential mediator of the RAS is aldosterone, which plays a crucial role in maintaining sodium and potassium balance, thus influencing extracellular volume. Aldosterone also exhibits potent pro-fibrotic effects [39]. The release of aldosterone is triggered by Ang II, and emerging evidence suggests that aldosterone may contribute to and exacerbate the detrimental effects of Ang II. Through activation of mineralocorticoid receptors, aldosterone promotes endothelial dysfunction and thrombosis, reduces vascular compliance and baroreceptor function, and induces fibrosis in both myocardial and vascular tissues [40]. The resulting increase in blood pressure and circulating volume, caused by the effects of Ang II and aldosterone on their target organs, establishes a negative feedback loop that suppresses renin release. The maintenance of this feedback inhibition critically relies on the Ang II-mediated activation of the AT1R [41].

Understanding the pathological effects of the RAS and the involvement of ROS is crucial for several reasons. It provides insights into the molecular mechanisms underlying cardiovascular diseases associated with RAS dysregulation, aiding in the advancement of precision therapeutic interventions. The identification of specific proteins and pathways involved opens opportunities for drug development, enabling the design of medications that selectively target these molecules or their associated receptors. This knowledge also contributes to personalized medicine approaches, considering individual variability in RAS responses. Moreover, it drives research advancements by uncovering novel signaling pathways and potential biomarkers, enhancing our understanding of cardiovascular disease pathophysiology. Overall, comprehending the pathological effects of RAS and ROS has significant implications for improving patient outcomes and advancing cardiovascular research.

### 2.3. The Application of ACEIs and ARBs in the Treatment of Cardiovascular Disorders

For a considerable period, the ACE/Ang II/AT1R pathway was recognized as the primary mechanism through which the RAS influences cardiovascular processes. Various categories of antihypertensive medications, including ACEIs, ARBs, β blockers (BBs), direct renin inhibitors (DRIs), and mineralocorticoid receptor antagonists (MRAs), have been employed to provide cardiorenal protection. ACEIs and ARBs, along with BBs, DRIs, and MRAs, act by interfering with the signaling pathways within the RAS. They are considered the first-line treatment options for managing hypertension and other cardiovascular disorders such as heart failure [42,43,44]. Furthermore, ACEIs and ARBs have a significant impact on the cardiovascular system and offer a protective effect against the occurrence and progression of kidney disease [45].

In addition to the well-established protection that ACEIs offer to the cardiovascular and renal systems by effectively controlling arterial pressure thresholds, they have also been associated with additional beneficial effects on the endothelium [46]. This has been confirmed in research studies conducted on patients with coronary artery disease or hypercholesterolemia, regardless of the effect on blood pressure reduction. These studies demonstrate that ACEIs and ARBs can additionally improve endothelial and vascular function in these patients. Furthermore, large studies have shown that ACEIs and ARBs are effective as monotherapy in managing other conditions such as left ventricular hypertrophy, systolic dysfunction, heart failure, and myocardial infarction [47]. Additionally, trials like the HOPE study, ONTARGET, and TRANSCEND have reported the extended benefits of ACEIs and ARBs to patients with an increased susceptibility to adverse outcomes but without left ventricular dysfunction [48]. For instance, the HOPE trial showed that ramipril significantly reduced the incidence of death, myocardial infarction, and stroke in high-risk patients [49]. Similarly, the EUROPA trial revealed a relative risk reduction in cardiovascular events with perindopril treatment in patients with stable coronary heart disease. However, the PEACE trial, which focused on stable coronary artery disease patients with preserved ventricular function, did not find a therapeutic benefit of ACEIs when added to conventional therapy, potentially because of the minimal occurrence of significant outcome events in that patient group [47,50].

The findings presented in this section contribute to our understanding of the significant benefits offered by ACEIs and ARBs in the realm of cardiovascular disease. These medications not only effectively lower blood pressure but also exhibit additional cardioprotective and reno-protective effects. The evidence highlights their ability to improve endothelial function, enhance vascular structure, and ultimately lead to improved cardiovascular outcomes. ACEIs and ARBs are essential primary therapies for high blood pressure, cardiac dysfunction, heart attack, and chronic kidney disease. Their therapeutic impact extends to high-risk patients without left ventricular dysfunction, further underscoring their clinical relevance. The comprehensive knowledge gained from large-scale randomized clinical trials supports the utilization of ACEIs and ARBs in the management of cardiovascular conditions, promoting improved patient outcomes and cardiovascular health.

### 2.4. Enhancing Endothelial Function and Mitigating Oxidative Stress: Exploring the Effects of ACEIs and ARBs

Several studies have demonstrated the favourable effects of ACEIs and ARBs on endothelial function and their ability to control oxidative stress levels in populations with cardiovascular diseases. As it has already been cleared, Ang II excessive presence is thought to play a pivotal role in the increased generation of O_2_^•_^ and the impairment of endothelial function in blood vessels. This effect is achieved through the activation of NADPH oxidases, triggered by the stimulation of AT1 receptors [51]. Additionally, studies conducted in laboratory settings have revealed that AT1 receptors can be upregulated by low-density lipoproteins [52,53]. By augmenting the activity of SOD3 and blocking NADPH oxidase activation, ACEIs and ARBs provide an indirect mitigation of oxidative stress [46,54]. In relation to ACEIs, clinical trials have demonstrated favourable outcomes in terms of improving endothelial function in individuals with hypertension or cardiovascular history. Yet, there remains a scarcity of research investigating the impact of ACEI treatment regarding the decrease in C-Reactive Protein (CRP) levels, highlighting a need for further exploration in this area [55,56,57,58,59,60] (Table 1). Additionally, while not the primary focus of this review, it is worth mentioning that research conducted in experimental studies indicates that ACEIs and ARBs may offer advantages in preventing or slowing down cognitive decline and dementia [61].

In patients with coronary artery disease, it has been observed that ramipril (ACEI) and losartan (ARB) improve endothelial function by increasing the availability of nitric oxide (NO) through the mitigation of oxidative stress within the arterial wall [62]. Additionally, ARBs offer vascular protective effects [63]. Losartan promotes the phosphorylation of endothelial NO synthase (eNOS) and suppresses endothelial apoptosis induced by Tissue Necrosis Factor-α (TNF-α) through the activation of the VEGFR2/PI3K/Akt pathway [61]. Additionally, in diabetic rats, losartan restored glomerular NO production by increasing GCH1 protein expression and tetrahydrobiopterin (BH4) levels [64]. Furthermore, valsartan and irbesartan (ARBs) exert effects that counteract the development of atherosclerosis. The promotion of eNOS Ser117-phosphorylation increases eNOS mRNA stability and this leads to the reduced NADPH oxidase expression and also to the augmented vascular BH4 and restored eNOS uncoupling [65,66]. More recent studies have revealed significant suppression of NADPH oxidase p22(phox) expression in the aortic wall of patients with thoracic aortic aneurysm following ARB treatment, which the authors attribute to the pleiotropic effects of ARBs on vascular metabolism [67].

Some of the positive effects of ACEIs may be attributed to their impact on an ACE signaling cascade, resulting in improved endothelial function that appears to be independent of their effects on vasoactive substances [68]. Another mechanism suggested to beneficially affect the endothelium is the introduction of ACEIs as a treatment, with potential ACE signaling cascade involvement. This effect seems unrelated to the impact of ACEIs on vasoactive substances [69]. Moreover, when ACEI binds to the cell surface ectoenzyme ACE, it triggers a cascade that ultimately increases ACE and cyclooxygenase-2 expression. Firstly, ACEI binding activates casein kinase 2, leading to serine residue phosphorylation at the molecule’s C-terminal end [69]. Secondly, ACE-associated c-Jun N-terminal kinase is activated, possibly through mitogen-activated protein kinase 7 activation [70]. This cascade eventually leads to an increase in the expression of ACE and cyclooxygenase-2 (through the accumulation of phosphorylated c-Jun in the nucleus, enhancing the DNA-binding activity of activator protein-1) [71]. Additionally, the elevated expression of cyclooxygenase-2 benefits endothelial function by promoting the production of prostacyclin, a vasodilator and antiplatelet agent, by endothelial cells [72].

Fleming et al. [71] have elucidated another aspect of ACE’s outside-in signaling role. According to their study, when an ACE inhibitor interacts with ACE, it triggers a series of signaling events that affect the synthesis of multiple proteins. This implies that ACEIs exert their beneficial effects through the activation of a unique signaling cascade mediated by ACE, which goes beyond mere alterations in Ang II and bradykinin levels. Another important signaling of ACEIs is the signaling of kinin B1 and B2 receptors (B1R, B2R). B2R functions and ACEIs can act similarly as allosteric agonists. B1R and B2R signaling enhancement help ACEIs in promoting the production of NO, a major contributor to many cardiovascular conditions’ treatments. Additionally, ACEIs have been shown to decrease CD40L levels, a protein predominantly present in activated T cells, as evidenced by well-established research [73,74]. Furthermore, recent studies have demonstrated that ACEIs also elevate adiponectin levels, a hormone responsible for regulating diverse metabolic processes [75,76].

Several studies have explored the shared effects of ACEIs in terms of their anti-hypertensive and anti-inflammatory actions. In one particular study, the ACEI captopril, which contains a sulfhydryl group, was compared to enalapril, an ACEI without a sulfhydryl group [77]. The findings suggested that ACEIs with a sulfhydryl group, such as captopril, possess the ability to protect the vascular endothelium against damages induced by L-methionine. The beneficial effects of captopril were associated with the attenuation in decreased activity of Paraoxonase-1 and NO levels. Another study investigated the role of the sulfhydryl group by comparing zofenopril and lisinopril in rats with myocardial infarction-induced heart failure. The results aligned with the previous study, indicating that the presence of a sulfhydryl group may offer a potential advantage in improving endothelial dysfunction through increased NO activity released from the endothelium into the vessel wall [78].

The concept of achieving more comprehensive inhibition of RAS and overcoming the “ACE escape” phenomenon led to the development of ARBs. ARBs effectively block the harmful effects of Ang II at the AT1 receptor, but they may diminish the beneficial effects of kinins. Interestingly, it was discovered that increased levels of kinins, previously considered an undesirable consequence of ACEIs, promote vasodilation and offer benefits. ARB therapy can activate the AT2 receptor, which potentially results in favourable anti-inflammatory, antithrombotic, and antiproliferative effects. Furthermore, losartan blockade of AT1R increases Ang II metabolism to angiotensin IV, and thus increases AT4R activation [79]. Some studies have reported relatively lower rates of myocardial infarction with non-ARB antihypertensive treatments compared to ARB-based treatments, leading to the “ARB-myocardial infarction paradox”. It has been suggested that this paradox arises from the unopposed activation of the AT2 receptor [79,80]. Although these receptors are generally associated with favourable effects, studies in certain animal models have shown that they can have hypertrophic and pro-inflammatory effects [79]. For instance, mice lacking the AT2 receptor are protected against cardiac hypertrophy, while overexpression of the AT2 receptor in isolated human cardiomyocytes is linked to hypertrophy [81]. Moreover, AT2 receptor activation has been found to stimulate the production of matrix metalloproteinase-1, an enzyme involved in degrading the fibrous cap of atherosclerotic plaques [82]. Therefore, AT2 receptor activation could contribute to plaque instability and the formation of blood clots [80].

Furthermore, ACEIs and ARBs contribute to the progression of atherosclerosis and oxidative stress by decreasing asymmetric dimethylarginine (ADMA) levels through the stimulation of dimethylarginine dimethylaminohydrolase (DDAH) activity [76,83]. It is well established that decreased ADMA levels enhance the coupling of eNOS. Nevertheless, the precise mechanism underlying the influence of RAS inhibitors on ADMA metabolism remains uncertain. Ang II fosters the generation of ROS via vascular NADPH oxidase [84,85,86,87]. The inactivation of DDAH by ROS suggests that ACEIs and ARBs may improve ADMA metabolism by reducing oxidative stress. Indeed, certain studies have shown that these drugs can reduce serum markers of oxidative stress [88,89]. For example, a study where the impacts of two ACE inhibitors were compared [90] found that zofenopril, which includes reduced sulfhydryl groups and displays direct antioxidant characteristics, and enalapril, which lacks -SH groups and does not demonstrate antioxidant activity. Zofenopril was more effective in reducing ADMA concentration. However, other studies [91] have not observed changes in serum lipid peroxidation products in patients treated with ACEIs, indicating that additional mechanisms should be considered.

The decrease in ADMA levels resulting from RAS blockade could be attributed to the reduction in blood pressure, as shear stress increases ADMA production by endothelial cells [92]. Therefore, lowering blood pressure through RAS blockade could lead to a decrease in ADMA [92,93]. Some studies have reported a simultaneous decrease in both ADMA and blood pressure, supporting this possibility [93]. However, other studies have shown that ACEIs or ARBs decrease ADMA levels without affecting blood pressure [89]. Additionally, the observation that only perindopril, but not bisoprolol, reduces ADMA in hypertensive patients despite a similar decrease in blood pressure suggests that the effect on blood pressure may not be the primary factor. Most studies involving ACEIs or ARBs have not reported changes in renal function, indicating that the reduction in ADMA concentration is unlikely to be solely due to improved renal excretion. In fact, in patients with type 2 diabetes, treatment with perindopril reduced plasma ADMA levels but had no effect on urinary ADMA levels [91].

To conclude this section, it is evident that ACEIs and ARBs have shown potential in modulating ADMA metabolism and improving endothelial function. These drugs may reduce ADMA levels through various mechanisms, including activation of DDAH and attenuation of oxidative stress. However, further research is required to fully elucidate the precise mechanisms involved. Understanding the impact of ACEIs and ARBs on ADMA metabolism contributes to our knowledge of their potential benefits in cardiovascular health and warrants continued investigation in this field.

## 3. Clinical Implications

The findings discussed in this paper have significant clinical implications for the use of ACEIs and ARBs in the management of cardiovascular diseases. The beneficial effects of ACEIs and ARBs extend beyond their traditional role as blood pressure-lowering agents. Firstly, ACEIs and ARBs have demonstrated their efficacy in improving endothelial function, which plays a crucial role in maintaining cardiovascular health. These drugs have been shown to increase the bioavailability of NO by reducing oxidative stress and enhancing NO production. Improved endothelial function is associated with vasodilation, anti-aggregatory effects, and reduced inflammation, ultimately leading to better cardiovascular outcomes. Therefore, ACEIs and ARBs should be considered as first-line therapies in patients with hypertension, coronary artery disease, and other cardiovascular disorders characterized by endothelial dysfunction.

Secondly, ACEIs and ARBs have shown potential in modulating ADMA metabolism. Elevated levels of ADMA, an endogenous inhibitor of NOS, have been linked to endothelial dysfunction and increased cardiovascular risk. ACEIs and ARBs may reduce ADMA levels through various mechanisms, including activation of DDAH and attenuation of oxidative stress. By improving ADMA metabolism and restoring NOS activity, these drugs can further enhance endothelial function and mitigate the progression of cardiovascular diseases.

Moreover, the pleiotropic effects of ACEIs and ARBs go beyond their role in blood pressure control. These drugs exhibit antioxidant properties, inhibit inflammatory pathways, and have potential anti-atherosclerotic effects. They may also modulate the RAS beyond blocking the effects of Ang II, leading to the activation of beneficial pathways such as the AT2 receptor. These broader effects contribute to their cardioprotective actions and suggest their potential utility in a range of cardiovascular disorders, including heart failure, myocardial infarction, and chronic kidney disease.

## 4. Limitations

While ACEIs and ARBs have demonstrated significant clinical benefits, it is important to acknowledge certain limitations associated with their use. These limitations should be taken into consideration when interpreting the findings and implementing these therapies in clinical practice.

Firstly, individual patient variability and heterogeneity in response to ACEIs and ARBs may impact their effectiveness. The response to these drugs can vary based on factors such as genetic variations, underlying comorbidities, and concomitant medications. Some patients may experience limited or suboptimal response to ACEIs or ARBs, necessitating the consideration of alternative treatment strategies or combination therapies. Secondly, although ACEIs and ARBs have shown promising effects on endothelial function, ADMA metabolism, and other cardiovascular parameters, the precise mechanisms underlying these effects are not fully understood. The complex interplay between RAS, oxidative stress, inflammatory pathways, and other factors involved in cardiovascular pathophysiology requires further investigation. A deeper understanding of these mechanisms could potentially help refine the use of ACEIs and ARBs and identify patient subgroups who would benefit the most from these therapies. Thirdly, like any medication, ACEIs and ARBs are associated with potential adverse effects. Common side effects include hypotension, hyperkalaemia, and renal dysfunction. Monitoring of blood pressure, electrolyte levels, and renal function is important during treatment. Additionally, individual patient characteristics and preferences should be considered when selecting the most appropriate therapy, as some patients may be more prone to specific side effects or have contraindications for ACEIs or ARBs.

Furthermore, the evidence supporting the clinical use of ACEIs and ARBs is primarily based on observational studies, randomized controlled trials, and meta-analyses. While these studies provide valuable insights, they also have inherent limitations, such as the potential for selection bias, confounding factors, and limited generalizability to diverse patient populations. Future well-designed clinical trials are needed to further validate the findings and establish the optimal use of these therapies in specific patient groups. Recognizing these limitations is crucial for clinicians to make informed treatment decisions and to individualize therapy for optimal patient outcomes. Further research is needed to address these limitations and expand our knowledge on the use of ACEIs and ARBs in diverse patient populations and clinical scenarios.

## 5. Conclusions

In summary, this paper emphasizes the paramount significance of the RAS in the genesis and advancement of hypertension, cardiovascular diseases, and renal disorders. The RAS has been subject to extensive scrutiny and its role in these conditions has been firmly established. Additionally, this study sheds light on the vital influence of ROS in maintaining the balance within the blood vessels and their intricate interactions with the RAS, impacting the course of diseases. Notably, therapeutic interventions aimed at the RAS, such as ACEIs and ARBs, have demonstrated beneficial effects on endothelial function and oxidative stress levels. Furthermore, ongoing research is delving into the underlying mechanisms governing the interplay between the RAS and ROS, offering new insights and potential innovative therapeutic avenues. Gaining a comprehensive understanding of these complex interactions is pivotal for the development of targeted and efficacious treatments for cardiovascular conditions.

## Figures and Tables

**Figure 1 ijms-24-12111-f001:**
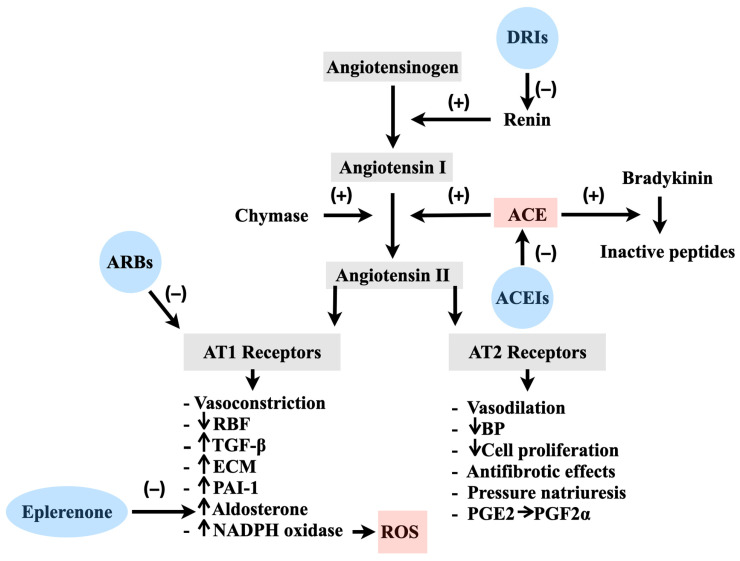
Interactions of the Renin-Angiotensin System, Kinin System, and Vascular Redox State: Implications for ACEIs and ARBs. ACE: Angiotensin-converting enzyme; AT1: Angiotensin type 1; AT2: Angiotensin type 2; ACEIs: ACE inhibitors; ARBs: AT1 receptor blockers; BP: Blood pressure; DRIs: Direct renin inhibitors; ECM: Extracellular matrix; NADPH: Nicotinamide adenine dinucleotide phosphate; PAI^_^1: Plasminogen activator inhibitor^_^1; PG: prostaglandin; RBF: Renal blood flow; ROS: Reactive oxygen species; TGF^_^β: Transforming growth factor^_^β.

**Table 1 ijms-24-12111-t001:** Randomized controlled trials assessing the influence of ACEIs and ARBs on serum CRP levels and endothelial function.

Author	Agents	Population	Period	Outcome
Walczak-Galezewska et al. [54]	Ramipril vs. Nebivolol	Hypertension	12 weeks	Ramipril demonstrated a decrease in CRP levels in comparison to nebivolol.
Ridker et al. [55]	HCTZ/Valsartan vs. Valsartan	Hypertension	6 weeks	CRP levels were reduced solely through treatment with valsartan.
Ghiadoni et al. [56]	Lisopril	Ventricular hypertrophy	3 years	Endothelial function showed improvement with lisinopril treatment when compared to the initial baseline.
Ghiadoni et al. [57]	Candesartan	Hypertension	1 year	Endothelial function demonstrated improvement with candesartan treatment when compared to the initial baseline.
Schiffrin et al. [58]	Losartan	Hypertension	1 year	Endothelial function showed improvement with losartan treatment in comparison to the initial baseline.
Schmieder et al. [59]	Telmisaran	Hypertension and DM	9 weeks	Endothelial function demonstrated improvement with telmisartan treatment in comparison to the initial baseline.

CRP: C-Reactive Protein; HCTZ: Hydrochlorothiazide.

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
