# Peer review of "Angiotensin Regulation of Vascular Homeostasis: Exploring the Role of ROS and RAS Blockers"

_ijms, 2023, doi:10.3390/ijms241512111_

Round 1

Reviewer 1 Report

 This manuscript is well-researched and well-written.  There is excellent coverage of the background literature and a pleasing absence of grammatical / syntactic errors.

I have three comments regarding either accuracy of the information presented, or further extension of that information:

Line 74:  It is my understanding that an AT3R has never been identified.  AT1R and AT2R were identified as targets for angiotensin II.  The AT4R was identified at the target for angiotensin IV.  Because of the numerical sequence of the nomenclature, researchers assumed, therefore that an AT3R had been identified and characterised.  Is this actually the case?

Lines 75-76:  There is much debate in the literature concerning the nature of the AT4R.  Probably the leading contender is insulin-regulated aminopeptidase (IRAP).  Angiotensin IV inhibits the peptidase activity of the enzyme, which is also called oxytocinase.  Angiotensin IV also promotes translocation of IRAP to the cell surface and promotes insulin-mediated glucose uptake.

Lines 281-282:  There is also evidence that losartan blockade of AT1R results in enhanced metabolism of angiotensin II to angiotensin IV, and thus increased AT4R activation.

This review concentrates on the cardiovascular disease and clinical implications but could go further, for example consideration of prevention of vascular dementia.  The effects of ACEis and ARBs of oxidative stress have also been considered in the literature concerning Alzheimer’s disease /dementia.  These drugs seem to have a protective effect against these conditions independent of the cardiovascular effects.  Although not the subject of this review, this fact is worthy of some mention.

Author Response

Dear Sir/Madam,

Thank you for your valuable comments. Your guidance has been immensely helpful. We have carefully considered your comments and made the necessary revisions accordingly

*I have three comments regarding either accuracy of the information presented, or further extension of that information:

-Line 74:  It is my understanding that an AT3R has never been identified.  AT1R and AT2R were identified as targets for angiotensin II.  The AT4R was identified at the target for angiotensin IV.  Because of the numerical sequence of the nomenclature, researchers assumed, therefore that an AT3R had been identified and characterised.  Is this actually the case?

Reply: We agree with the reviewer's opinion, and the text concerning these receptors has been corrected (Lines 91-92).

-Lines 75-76:  There is much debate in the literature concerning the nature of the AT4R.  Probably the leading contender is insulin-regulated aminopeptidase (IRAP).  Angiotensin IV inhibits the peptidase activity of the enzyme, which is also called oxytocinase.  Angiotensin IV also promotes translocation of IRAP to the cell surface and promotes insulin-mediated glucose uptake.

Reply: Thank you for this valuable information which has been added in Lines 94-97.

-Lines 281-282:  There is also evidence that losartan blockade of AT1R results in enhanced metabolism of angiotensin II to angiotensin IV, and thus increased AT4R activation.

Reply: Thank you for this valuable information which has been added in Lines 296-298.

*This review concentrates on the cardiovascular disease and clinical implications but could go further, for example consideration of prevention of vascular dementia.  The effects of ACEis and ARBs of oxidative stress have also been considered in the literature concerning Alzheimer’s disease /dementia.  These drugs seem to have a protective effect against these conditions independent of the cardiovascular effects.  Although not the subject of this review, this fact is worthy of some mention.

Reply: Thank you for providing this valuable information, which has been incorporated into the relevant sections of the text (Lines 228-231)

Reviewer 2 Report

The manuscript is well written and has useful information for readers. I just have several minor points to mention.

* Title is a little bit misleading. I expect the review will highly focus on ROS but it is actually a rather small part of the whole story. Maybe you can change the title to 'Angiotensin Regulation of Vascular Homeostasis: Exploring the role of ROS and RAS blockers'?

* The first paragraph in page 17 seems to be a repetition of the previous paragraph. 

* Can you put a reference for the sentence in page 17 line 309-311?

Author Response

Dear Madam/Sir,

Thank you for your valuable comments. Your guidance has been immensely helpful. We have carefully considered your comments and made the necessary revisions accordingly

* Title is a little bit misleading. I expect the review will highly focus on ROS but it is actually a rather small part of the whole story. Maybe you can change the title to 'Angiotensin Regulation of Vascular Homeostasis: Exploring the role of ROS and RAS blockers'?

Reply: Thanks for your valuable advice. The title has been changed.

* The first paragraph in page 17 seems to be a repetition of the previous paragraph. 

Reply: We have conducted a comprehensive rewording and reconstruction of our paper, and we believe that we have successfully addressed any potential issues of repetition.

* Can you put a reference for the sentence in page 17 line 309-311?

 Reply:  A reference was added for the sentence which now is in lines 326-328.